# Effects of Incretin-Based Treatment on the Diastolic (Dys)Function in Patients with Uncontrolled Type 2 Diabetes Mellitus: A Prospective Study with 1-Year Follow-Up

**DOI:** 10.3390/diagnostics13172817

**Published:** 2023-08-31

**Authors:** Elena-Daniela Grigorescu, Cristina-Mihaela Lăcătușu, Mariana Floria, Georgiana-Diana Cazac, Alina Onofriescu, Livia-Amira Sauciuc, Alexandr Ceasovschih, Ioana Crețu, Bogdan-Mircea Mihai, Laurențiu Șorodoc

**Affiliations:** 1Unit of Diabetes, Nutrition and Metabolic Diseases, Faculty of Medicine, “Grigore T. Popa” University of Medicine and Pharmacy, 700115 Iași, Romania; elena-daniela-gh-grigorescu@umfiasi.ro (E.-D.G.); alina.onofriescu@umfiasi.ro (A.O.); bogdan.mihai@umfiasi.ro (B.-M.M.); 2Clinical Center of Diabetes, Nutrition and Metabolic Diseases, “Sf. Spiridon” County Clinical Emergency Hospital, 700111 Iași, Romania; 3Department of Internal Medicine, Faculty of Medicine, “Grigore T. Popa” University of Medicine and Pharmacy, 700115 Iași, Romania; floria.mariana@umfiasi.ro (M.F.); alexandr.ceasovschih@umfiasi.ro (A.C.); laurentiu.sorodoc@umfiasi.ro (L.Ș.); 4Medical Clinic, “Sf. Spiridon” County Clinical Emergency Hospital, 700111 Iași, Romania; 5Faculty of Medicine, “Grigore T. Popa” University of Medicine and Pharmacy, 700115 Iași, Romania; mg-rom-31094@students.umfiasi.ro; 6Crețu R. Ioana PFA, 1 Mărului, 707020 Aroneanu, Romania; contact@ioanacretu.ro

**Keywords:** type 2 diabetes mellitus, diastolic dysfunction, DPP-4 inhibitors (sitagliptin/saxagliptin), GLP-1 receptor agonist (exenatide), prospective study

## Abstract

Left ventricular diastolic dysfunction (DD) is a subclinical cardiac abnormality in patients with type 2 diabetes mellitus (T2DM) that can progress to heart failure (HF) and increase cardiovascular risk. This prospective study evaluated the DD in T2DM patients without atherosclerotic cardiovascular disease after one year of incretin-based drugs added to standard treatment. Of the 138 enrolled patients (49.30% male, mean age 57.86 ± 8.82, mean T2DM history 5 years), 71 were started on dipeptidyl peptidase-4 inhibitor sitagliptin/saxagliptin, 21 on glucagon-like peptide-1 receptor agonist exenatide, and 46 formed the control group (metformin and sulphonylurea/acarbose). At baseline, 71 patients had grade 1 DD, another 12 had grade 2 and 3 DD, and 15 had indeterminate DD. After one year, DD was evidenced in 50 cases. Diastolic function improved in 9 cases, and 27 patients went from grade 1 to indeterminate DD. The active group benefited more, especially patients treated with exenatide; their metabolic and inflammation profiles also improved the most. An in-depth analysis of echocardiographic parameters and paraclinical results in the context of literature data justifies the conclusion that early assessment of diastolic function in T2DM patients is necessary and the benefits of affordable incretin-based treatment may extend to subclinical cardiovascular manifestations such as DD.

## 1. Introduction

### 1.1. Diabetes, Obesity, and Cardiovascular Disease—Global Context and Predicted Trends

Diabetes is widely recognized as a “global crisis” of growing proportions. The International Diabetes Federation (IDF) reported that, in 2021, over half a billion people were known to have diabetes and another three hundred million had impaired glucose tolerance. Close to a quarter billion people were believed unaware of having the disease. The annual global healthcare bill for diabetes-related treatments was estimated at almost a trillion dollars and, sadly, diabetes either caused or contributed significantly to the deaths of 6.7 million people that year. In Romania, close to 1.2 million adults had diabetes (more than 8% of the country’s adult population), and undiagnosed diabetes was estimated at more than 20% [1]. 

Three-quarters of patients with diabetes live in low- and middle-income countries, and close to half of them are residents of urban areas. These are also the places where prevalence is expected to rise most sharply. Societal and economic inequities are now recognized as powerful, structural environmental forces that inevitably thwart and overwhelm individual choices by facilitating unhealthy food addictions while concurrently limiting access to healthy choices. The evidence is unmistakably pointing to a grimmer, rather than brighter, future in the absence of deep, paradigmatic, systemic changes that address individual people in their contexts. By the end of this decade alone, the IDF predicts that an additional hundred million adults will be newly diagnosed with diabetes [1,2,3].

The pathophysiological processes that lead to, sustain, and aggravate diabetes have been thoroughly researched. Of the main culprits, obesity is front and center. Unsurprisingly, reports on obesity make similar predictions as the aforementioned projections regarding diabetes. This is because the contributing socio-economic issues are, essentially, the same. The World Obesity Federation (WOF) calculated that, in 2020, more than 2.6 billion people were already overweight, having body mass indices (BMIs) ≥ 25 kg/m^2^. It also estimated that over 4 billion people will have crossed this threshold by 2035; of them, as many as half are predicted to become clinically obese (BMI ≥ 30 kg/m^2^). For Romania, experts predict that 35% of the country’s adult population will be living with obesity by 2035 [4].

People with diabetes mellitus and with obesity have a substantially higher cardiovascular risk than the general population, ranging from moderate to very high. Strong evidence highlights that physical activity behaviors, healthy eating patterns, and weight management should be integrated into a holistic approach to diabetes management [5]. All types of physical activity, especially moderate to vigorous, improve cardiometabolic health in people with type 2 diabetes mellitus (T2DM). Levels of physical exertion that get the body to sweat have been shown to improve physical function, blood pressure, metabolic profile (glucose/insulin, HbA_1c_, lipids), as well as quality of life and wellbeing (preventing and alleviating depression). These benefits are also present when other types of diabetes are managed in this way, provided macronutrient intake recommendations take into consideration the type and intensity of physical exercise practiced [6].

Addressing individual nutrition needs is key in diabetes management in order to identify and promote healthy eating patterns protective of cardiovascular health [5]. Various eating patterns have been under scientific scrutiny with regard to their cardiometabolic benefits, such as the Mediterranean and vegetarian diets, but also intermittent fasting and even, more recently, ketogenic-diet-based interventions. For the latter, some authors reported body weight and metabolic improvements, as well as anti-inflammatory effects ameliorating cardiac function in overweight T2DM patients [7,8,9]. Epidemiological data from fifty-seven studies conducted during 2010–2017 showed that approximately one-third of T2DM adults already had established cardiovascular disease at that point. Despite widespread recognition of the importance of a multifactorial approach, future projections do not indicate a reduction of cardiovascular disease in people living with diabetes [10].

### 1.2. Diastolic Dysfunction in Type 2 Diabetes Mellitus—Benefits of Antihyperglycemic Medication

Left ventricular (LV) diastolic dysfunction (DD) is a subclinical cardiac abnormality that may develop in patients with diabetes mellitus and eventually progress to heart failure (HF), with potentially life-threatening consequences. It is easy to diagnose, and routine echocardiographic investigations are of undeniable value for the early detection of DD in patients with diabetes [11].

Preclinical diastolic dysfunction (DD) unfolds insidiously while systolic function is normal and in the absence of any symptoms of heart failure (what the American Heart Association defines as HF stage B) [12]. Preclinical DD is also associated with reduced quality of life and structural abnormalities that increase cardiovascular risk [13]. This calls for early intervention to slow down or even reverse any underlying DD, as one-third of patients with normal systolic function but DD may progress to HF [14].

The progression from DD to HF with preserved ejection fraction (EF) is more common in patients with hypertension (HTN), T2DM, coronary artery disease (CAD), and anemia, but the frequency of progression from the preclinical stage to clinically evident myocardial dysfunction is not well established [15,16]. In their review, Wan et al. cited studies that found moderate or severe DD in less than half of subjects with documented HF (Redfield), in contrast to Abhayaratna, who found DD in 86% of patients, 36% of whom were asymptomatic for HF according to New York Heart Association (NYHA) criteria [16]. Among the 23% of patients with T2DM and diagnosed DD based on the ratio of early diastolic mitral inflow velocity to early diastolic mitral annulus velocity (E/e’ ratio), the probability of developing HF after 5 years of follow-up was 36.9% compared to 16.8% in patients without DD [17].

The metabolic conditions and physiological modifications secondary to chronic hyperglycemia and hyperlipidemia form a cascade of interlinked pathological processes. DeFronzo’s ominous octet of contributors to glucose intolerance has recently been extended to as many as eleven, and medication is available to address each type of imbalance [18,19,20]. A research question of significant practical implications is to what extent antihyperglycemic drugs can slow the progression of insidious cardiovascular manifestations in diabetes patients without known atherosclerotic disease [21,22].

With regard to metformin, the prospective MET-DIME trial enrolled 54 non-diabetic patients with DD, providing either lifestyle counseling alone or metformin in addition to lifestyle counseling. The patients were monitored for 24 months; the results after 12 months indicated that 2000 mg of metformin daily had beneficial effects on the diastolic dysfunction parameters e’, the ratio of the early (E) to late (A) ventricular filling velocities (E/A), and isovolumetric relaxation time (IVRT), but only the increase of e’ was statistically significant [23]. In another study, the evaluation of diastolic function in 242 patients with T2DM revealed improved IVRT values in the group treated with metformin compared to patients receiving sulfonylurea or insulin-based treatment [24].

The effects of dipeptidyl peptidase-4 (DPP-4) inhibitors on diastolic function have not yet been investigated in longitudinal prospective studies on large cohorts. Available studies on smaller numbers of patients have so far associated sitagliptin with ameliorated diastolic dysfunction, but the methodologies employed were too heterogeneous to aggregate the results: different DD diagnostic and grading algorithms were used [22].

In a study on 35 uncontrolled T2DM patients treated with metformin and glibenclamide, the introduction of sitagliptin 100 mg/day led to decreased diastolic E/e’ indices after 24 weeks, independent of changes in blood pressure levels, as well as to improved systolic function in 46.7% of cases. These results were superior compared to a second group of patients whose treatments were changed to NPH insulin [14]. Similarly, Yamada et al. found a reduction in the E/e’ ratio after 2 years of treatment with sitagliptin compared with placebo in 115 patients with uncontrolled T2DM despite lifestyle changes and pharmacological treatment. This result was independent of changes in systolic function, blood pressure, and natriuretic peptide levels [25].

On the other hand, in a prospective randomized trial, Oe et al. reported no changes in e’ and E/e’ values after 24 weeks of treatment with either sitagliptin or voglibose added to pioglitazone in 80 patients with T2DM [26]. No significant changes in the parameters e’, E/A, and E/e’ were also noted after 48 weeks of linagliptin versus placebo in another study on 174 patients with T2DM, of whom 88 formed the experimental group and the other 86 were administered a placebo [27].

Other studies have reported on the effects of DPP-4 inhibitors (e.g., sitagliptin) relative to those of SGLT2 inhibitors such as empagliflozin. For instance, in a parallel group trial enrolling 44 Japanese patients with T2DM, echocardiographic changes were assessed after 12 weeks of treatment with either sitagliptin or empagliflozin. While LVEF and percentage fractional shortening (%FS) decreased significantly more in the sitagliptin group (*p* < 0.05), no significant differences were found regarding E/e’ and E/A [28].

In patients with established CVD, the cardiovascular benefits of sodium-glucose co-transporter 2 (SGLT2) inhibitors and glucagon-like peptide-1 receptor agonists (GLP-1RAs) have already been confirmed [29]. To fully elucidate the benefits of SGLT2 inhibition on diastolic (dys)function, large-scale clinical research with a substantial follow-up period would be needed in the way of the completed trials that have demonstrated the more overt cardiovascular outcomes. For now, smaller studies on T2DM patients with known CVD report promising results in terms of reduced E/e’ after 3 months of treatment with empagliflozin or with canagliflozin, or after 6 months of dapagliflozin [30,31,32]. Additionally, tofogliflozin, an SGLT2 inhibitor approved for the treatment of T2DM in Japan, was found to decrease E/e’ and improve diastolic function in 162 patients [33].

Another class of incretin-based treatment with beneficial effects on diastolic function are the GLP-1RAs, such as liraglutide. In a study on 49 T2DM patients without established CVD, of whom 23 were given liraglutide and 26 placebos, a regression of diastolic dysfunction severity was observed after 6 months of treatment with 1.8 mg liraglutide daily compared to placebo. The results included increased e’ and decreased left ventricular end-diastolic volume (LVEDV) [34]. Similarly, a reduction in the E/e’ ratio was noted in 37 T2DM patients without previous coronary artery disease, treated with liraglutide for 6 months [35].

Debate is ongoing regarding the underlying mechanism(s) for all these beneficial effects on diastolic and systolic cardiac function, specifically whether glucose lowering is the main factor or other, non-glycemic processes are independently involved. Several recent meta-analyses establish with sufficient data that GLP-1RAs are instrumental for patients with long histories of diabetes and associated comorbidities, but further research is needed to demonstrate if these drugs act directly or via glucose lowering [36]. The importance of non-glycemic effects is supported by evidence of reduced incidence of CVD after relatively few years of treatment, as shown in the HARMONY, LEADER, REWIND, and SUSTAIN-6 trials [37,38,39,40,41].

Apart from their established antihyperglycemic effects, GLP-1RAs reduce atherosclerotic plaque by inhibiting systemic inflammation, a process which is determined by decreasing leukocyte adhesion and extravasation, as well as decreasing extracellular matrix proteins, independent of the effects on lipid metabolism [42,43]. In clinical studies, administration of liraglutide and exenatide resulted in reductions of TNF-α, interleukins 1β and 6, CRP, and leukocyte adhesion molecules [44]. Liraglutide was found to reduce the macrophage activation marker sCD163, preventing atherosclerosis, and exenatide was shown to exert an anti-oxidative/anti-inflammatory effect [45].

The authors of a recent review analyzed the effects of antihyperglycemic drugs on cardiac function in patients with T2DM through their effects on endothelial function, lipid profile, blood pressure levels, and pro-inflammatory status, and showed a correlation between their benefits and reduced cardiovascular risk, but did not find sufficiently compelling evidence for the effects of GLP-1RAs on inflammation [45].

### 1.3. Study Aims

This study aimed to assess the effects of incretin-based treatment on the diastolic (dys)function of uncontrolled T2DM patients with no clinical signs of atherosclerotic cardiovascular disease. The two classes of incretin-based medication taken into consideration were GLP-1 receptor agonists (exenatide) and DPP-4 inhibitors (sitagliptin and saxagliptin) added to standard metformin treatment.

## 2. Results

The study enrolled 138 participants from a total of 639 patients presenting consecutively during the studied period. In total, 74 declined participation and 305 patients had known atherosclerotic cardiovascular disease (193 subjects) or were smokers. The clinical and biological evaluation led to the exclusion of another 115 patients: 47 had uncontrolled blood pressure, ischemic changes were detected on the electrocardiographic route in another 19 subjects, and 34 had triglyceride values above 400 mg/dL (see Figure 1).

The active group consisted of 92 patients, of whom 71 were started on sitagliptin/saxagliptin, while exenatide was introduced in the treatment of the other 21 patients. From the control group, 46 patients were treated with sulphonylurea and/or acarbose in addition to metformin. At the one-year follow-up, 132 patients were reevaluated (6 persons no longer met the inclusion criteria or did not respond to the invitation).

### 2.1. General Demographic and Clinical Characteristics

The mean age of the patients included in the study was 57.86 ± 8.82 years and 49.30% of the participants were male. The median history of T2DM was five years. With the exception of 12 patients who did not have comorbidities, 75.46% of participants had non-alcoholic fatty liver disease, 71.74% had dyslipidemia, and 67.39% were under antihypertensive treatment. Additionally, peripheral sensorimotor neuropathy was present in 44.2% of cases, and mild diabetic retinopathy in 10 patients (7.2%). Regarding excess weight, approximately two-thirds of the patients (66%) had BMI ≥ 30 kg/m^2^ and abnormal abdominal circumference was noted in all but three cases (109.13 ± 10.74 cm).

Table 1 provides a summary of the general clinical characteristics of the patients grouped according to the treatment administered for 52 weeks. At the beginning of the study, patients in both groups were homogenously distributed, as indicated by the comparative analysis of the clinical-biological data.

No significant differences were noted between the groups in terms of waist circumference and BMI indices at baseline and at follow-up. The mean BMI of patients treated with incretin-based drugs did increase, but the change +0.89 (±15.55) kg/m^2^ was not significant (*p* = 0.591). On the other hand, the patients in the incretin-based treatment group lost −2.19 ± 5.37 kg on average, which is statistically significant relative to the other group (*p* < 0.001).

Both treatment approaches led to significant improvement of glycemic control expressed as HbA_1c_ values (−0.66% vs. −0.45%, *p* < 0.001), while fasting glycemia levels decreased significantly after one year only in the incretin-based treatment group (165 vs. 148 mg/dL, *p* < 0.001). Regarding the insulin-resistance markers, HOMA based on fasting insulin or C-peptide decreased in both groups, but the differences were not significant.

The levels of inflammatory markers were similar in the two groups at baseline. At the end of the studied period, hsCRP was significantly lower in the incretin-based treatment group (*p* < 0.01). Additionally, IL-6 and TNF-α appeared to be increasing (significantly in the case of the latter, as measured in the active group).

The lipid profile at baseline featured abnormal mean levels of LDL cholesterol and triglycerides in both groups. After one year, no significant changes were noted, except for a significant reduction in high-density lipoprotein (HDL) cholesterol in the control group (−4.60 ± 14.27, *p* < 0.05). Regarding renal function, by the end of the study period, the eGFR and UACR had decreased significantly in patients treated with incretin-based agents (*p* < 0.01).

### 2.2. Echocardiographic Data and Assessment of Diastolic (Dys)Function

The echocardiographic assessment of diastolic function in patients is illustrated in Figure 2 and Figure 3. Only 29% of patients with normal ejection fraction (FEVS = 67.14%) presented normal diastolic function. The grade of severity could be established in 83 cases, while 15 were considered indeterminate. Of the patients in whom diastolic dysfunction (DD) could be graded, seventy-one patients (51.4%) had grade 1 DD, five patients had grade 2 DD, and seven had grade 3 DD. Comparisons between mean values of echocardiographic parameters measured to evaluate diastolic dysfunction (summarized in Table 2) confirmed the homogenous distribution of patients in the two groups relative to their initial cardiac status.

At baseline, the correlation analysis between the diastolic dysfunction parameters and other characteristics, respectively, revealed statistically significant positive associations between EDT and age (r = 0.237, *p* = 0.015), as well as between E/e’ and age (r = 0.327, *p* = 0.01). Negative significant associations were identified between E/A and age (r = −0.283, *p* = 0.003), respectively, between E/A and HDL cholesterol (r = −0.191, *p* = 0.049). Additionally, E/e’ was in a weak positive correlation with BMI, insulin, and HOMA-IR with r < 0.3 (*p* = 0.02 in all associations). Weak inverse correlations were found between EDT and hsCRP (−0.23, *p* = 0.01), as well as between LAVi and HbA_1c_ (−0.19, *p* = 0.03).

After one year, 48 patients presented normal diastolic function, while DD was found in 50 cases. Relative to baseline, at the end of the studied period, diastolic function was improved in 9 cases. Another 27 patients no longer presented grade 1 diastolic dysfunction; instead, their cardiac parameters suggested indeterminate DD.

There were no significant differences between the two groups at baseline or after one year in terms of LV ejection fraction, LAVi, LA aria, E/A, and e’. However, at follow up, EDT, E/e’, s’ lateral, IVS, and LVPW were significantly higher in the control group. In patients treated with incretin-based agents, small modifications in LV end diastolic and systolic diameters and IVRT were noted (*p* < 0.05).

### 2.3. Subgroup Analysis Comparing Outcomes Based on Each Incretin-Based Drug Administered

On enrollment, the incretin-based medication available through national insurance programs were exenatide from the class of GLP-1RAs, and either sitagliptin or saxagliptin from the class of DPP-4 inhibitors. In our active group, twenty-one patients were prescribed exenatide, sixty-four sitagliptin, and seven saxagliptin. When probing the data based on each medication class administered, we noticed several interesting significant differences. In what follows, the patients in the active group are referred to as two subgroups (exenatide and sitagliptin/saxagliptin).

In the exenatide-treatment subgroup, a more substantial, statistically significant re-duction of glycated hemoglobin levels was noted (z = −3.51, mean = 2.52%, *p* < 0.001) compared to the other subgroup (z = −4.74, mean = 0.52%, *p* < 0.001), as well as compared to patients in the control group (z = −3.26, mean = 0.56%, *p* = 0.007). The assessment of fasting glycemia at follow-up revealed statistically significant decreases in the overall active group (*p* < 0.001), but the greatest improvement occurred in the exenatide treatment subgroup (−50 mg/dl, z = −3.44, *p* = 0.001), followed by the DPP-4 inhibitors subgroup (−13.81 mg/dL, z = −2.57, *p* = 0.01), compared to −9.81 mg/dL, *p* = 0.137 in the control group.

Last but not least, in the exenatide-treatment subgroup, we also noted significant reductions in weight (−6.16 kg, z = 3.24, *p* = 0.001), BMI (−2.71 kg/m^2^, z = −3.29, *p* = 0.001), WC (−7.11 cm, z = −3.74, *p* < 0.001), HOMA-IR (1.95, z = −2.49, *p* = 0.013), and hsCRP (−4.5 mg/L, z = −2.43, *p* = 0.015).

With regard to the addition of sitagliptin to metformin treatment in the case of sixty-four patients from the active group, this had a beneficial effect on the metabolic and inflammation markers, while appearing to be neutral and safe for cardiac function. Only IVRT, left atrium diameter, and LVEDV changed significantly (*p* = 0.019, *p* = 0.012, and *p* = 0.035, respectively). The mean values of EF, e’, and LAVi decreased, while IVS, EDT, LVPW, and E/e’ increased, but these changes were not statistically significant.

Regarding metabolic and inflammation parameters, substantial improvement was noted for HbA_1c_ (7.38 ± 0.69%) and hsCRP (5.13 ± 4.82 mg/L) at *p* < 0.001, and also for HOMA-IR (4.46 ± 3.05) and fasting glycemia (159.44 ± 34.22 mg/dL) at *p* < 0.05.

### 2.4. Subgroup Analysis Based on the Patients’ Diastolic Function Assessments

The strengths of associations between metabolic markers, inflammation markers, and diastolic function parameters were also studied relative to the patients’ diastolic function assessments on enrollment. Thus, three subgroups were defined: absent DD (normal function), present DD (grade 1–3), and indeterminate DD. These results are summarized in Table 3.

The results in Table 3 help highlight that the relationship between age and several echocardiographic parameters is statistically significant in patients with present DD at baseline. Of the anthropometric parameters, positive correlations with weak-to-moderate strength were noted between BMI and E (*p* < 0.05), and between waist circumference and e’ (*p* < 0.05). From the insulin resistance markers, HOMA-IR and HOMA C-peptide were in a significant linear relationship with E values. Additionally, HOMA C-peptide appeared inversely correlated with LAVi in patients with DD. Regarding inflammation markers, hsCRP was in a significant negative association with EDT only (*p* < 0.05).

Additionally, the univariate analysis pointed to age as an independent risk factor for diastolic dysfunction (Exp B = 1.067, 95% CI = 1.014–1.122, *p* = 0.012). Importantly, a reduction of 1% in glycated hemoglobin levels decreased the risk of LVDD by 32%. The other studied markers, including diabetes duration, and known comorbidities or complications (HTA, neuropathy) did not present significant predictive value for LVDD.

After 12 months, the correlation analysis showed significant negative associations with moderate strength between age and diastolic index e’ (r = −0.420, *p* < 0.001), and between age and E/A (r = −0.429, *p* = 0.001). For the other echocardiographic parameters, positive correlations of weak to moderate strength were noted between age and E/e’ (r = 0.225, *p* = 0.009), LAVi (r = 0.270, *p* = 0.002), and EDT (r = 0.407, *p* < 0.001). The assessment of the relationship between age and E/A in each subgroup depending on DD status (absent/present/indeterminate) revealed a stronger negative association in the cases of indeterminate DD (r = −0.673, *p* < 0.001) compared to absent DD (r = −0.261, *p* = 0.037) and present DD (r = −0.297, *p* = 0.018). The strength of direct associations between age and the echocardiographic parameter EDT depending on DD status seemed to follow a similar pattern (r = 0.513 at *p* = 0.001 in the indeterminate DD subgroup vs. r = 0.364 at *p* = 0.005 in patients with graded DD).

Certain diabetes-related metabolic markers at the 1-year follow-up were significantly associated with the echocardiographic DD results. Glycated hemoglobin levels correlated negatively with EDT in patients with normal diastolic function (r = −0.277, *p* = 0.033), and positively with LAVi in patients with DD (r = 0.307, *p* = 0.018) and with IVRT in the subgroup with indeterminate DD (r = 0.306, *p* = 0.045). Additionally, a positive relationship was noted between the C-peptide index and IVRT (r = 0.179, *p* = 0.040) for all participants. Upon closer inspection, this appeared to originate from the indeterminate DD subgroup (r = 0.290, *p* = 0.048), where the C peptide index also correlated with LAVi (r = 0.249, *p* = 0.041). The insulin resistance marker HOMA-IR was associated with IVRT (r = 0.313, *p* = 0.014), LAVi (r = 0.274, *p* = 0.027), and E/A (r = −0.253, *p* = 0.041) in patients with normal diastolic function (absent DD). For patients with DD, the only noteworthy relationship was between HOMA-IR and LAVi (r = 0.274, *p* = 0.027).

Significant yet weak associations were also noted between certain inflammation markers and echocardiographic DD parameters at follow-up. These mostly involved TNF-α with E/e’ (r = 0.283, *p* = 0.001), E/A (r = −0.202, *p* = 0.021), and e’ (r = −0.272, *p* = 0.002). The other inflammation markers appeared to interact less with DD parameters when considering the entire study group. When considering DD status, it was again TNF-α which correlated with a greater number of echocardiographic parameters: e’ and E/e’ in the absence of DD (r = −0.412 at *p* = 0.002 and r = 0.385 at *p* = 0.003, respectively), e’ and EDT in the presence of DD (r = −0.238 at *p* = 0.04 and r = 0.346 at *p* = 0.008, respectively), and E/A in cases of indeterminate DD (r = −0.365, *p* = 0.035). For the subgroup of patients with DD, specifically, a pattern of weak significant associations emerged involving e’ and the full set of studied inflammation markers TNF-alfa (r = −0.238, *p* = 0.04), IL-6 (r = 0.255, *p* = 0.037), and hsCRP (r = 0.256, *p* = 0.038).

## 3. Discussion

This study evaluated the outcomes of incretin-based treatment (exenatide or sitagliptin/saxagliptin) added to standard antihyperglycemic therapy in adult patients with uncontrolled type 2 diabetes mellitus without atherosclerotic manifestations. The studied parameters describe the patients’ diastolic (dys)function according to baseline and follow-up echocardiographic investigations, but also their glycemic, metabolic, and inflammatory status, as indicated by laboratory tests.

### 3.1. Diabetes-Related Results, Metabolic and Inflammatory Status

The 138 patients enrolled in the study (49.30% male, mean age 57.86 ± 8.82) had an average history of established diabetes of 6.16 years. At baseline, the patients presented poor glycemic control (median HbA_1c_ 7.8%), which justified the clinical decision to enhance their standard antihyperglycemic treatment with incretin-based drugs (either exenatide or sitagliptin/saxagliptin). The patients also fitted the metabolic syndrome typology characterized by insulin resistance, hyperinsulinemia, increased visceral adiposity, hypertriglyceridemia, chronic subclinical inflammation, and hepatic steatosis (data reported previously) [46].

After one year of enhanced treatment, repeat tests and investigations revealed several significantly different metabolic parameters and inflammatory markers compared to patients who continued with standard treatment (the control group): weight loss, lower levels of total cholesterol, LDL cholesterol, triglycerides, uric acid, urine albumin–creatinine ratio, HbA_1c_, insulin resistance markers, hsCRP, and increased levels of HDL cholesterol, TNF-α, and IL-6.

The improvement of glycemic control in the active group of patients is similar to that reported in the literature. On enrollment, exenatide was available through the national Romanian health insurance program as one of the GLP-1RAs that could be administered weekly or twice a day. Currently, the drugs in this class have been ranked in terms of demonstrated antihyperglycemic effect intensity: semaglutide, followed by dulaglutide and liraglutide, then exenatide with prolonged release, and, as the least potent, exenatide administered daily and lixisenatide [47,48].

Regarding the patients’ overweight status, the average weight reduction in the active group was roughly 2 kg over the course of the studied year. The comparative analysis between the two groups revealed that patients who had been treated with exenatide were able to lose significantly more weight (approximately 6 kg, *p* = 0.001) and reduce their abdominal circumference substantially (approximately 7 cm, *p* < 0.001). The GLP-1 RA exenatide is resistant to the degradation exerted by the DPP-4 enzyme. Exenatide reduces fasting and postprandial hyperglycemia by increasing insulin secretion in a glucose-dependent manner, while also suppressing excess glucagon production. It also slows gastric emptying, diminishing the appetite, and increasing satiety, which helps patients reduce their food intake and achieve weight loss [49]. Delayed stomach emptying in patients treated with GLP-1 RAs for diabetes or weight management could require precautions in specific occasions, such as prior to undergoing anesthesia for surgical interventions, to avoid the risk of pulmonary aspiration of gastric contents [50]. In the DURATION-7 study, exenatide added to basal insulin effected a 1 kg weight reduction compared to patients given placebo, who actually gained weight (0.46 kg, *p* < 0.001) [51]. Additionally, in our study, compared to the control group, HOMA-IR decreased significantly under incretin-based treatment, the greatest reduction (−2) being noted in the subgroup of patients treated with exenatide.

From the class of DPP-4 inhibitors, sitagliptin and saxagliptin were the two options at baseline that did not incur additional costs for the patients (vildagliptin was introduced in the national insurance program in Romania after the study was completed). After one year of treatment with sitagliptin as an add-on to metformin, a reduction of 0.5% in HbA_1c_ was recorded (*p* < 0.001), similar to that in the PROLOGUE trial. In a subanalysis of these trial data, the achievement of glycemic control (HbA_1c_ ≤ 6.2%) after 24 months was not statistically significant, and other improvements in LDL cholesterol and hsCRP were equally non-significant [25].

As for inflammation, the marker hsCRP was found significantly decreased in the active group after 12 months (−3.6 mg/L under treatment with exenatide, *p* = 0.015, and −3.84 mg/L under treatment with sitagliptin, respectively). The meta-analysis from a recent systematic review of seven clinical trials showed that exenatide could reduce plasma hsCRP levels more than metformin, sulfonylurea, or insulin glargine [52]. In patients with T2DM, Satoh-Asahara reported a reduction of IL-6 and TNF-α simultaneous with an increase of GLP-1 and IL-10 under treatment with sitagliptin, compared to the control group [53].

### 3.2. Diastolic (Dys)Function and Cardiovascular-Related Results

The assessment of diastolic (dys)function in the present research was performed according to the algorithm for diagnosing diastolic dysfunction in patients with normal EF. At baseline, the prevalence of grade 1 diastolic dysfunction was approximately 50%, which is within the range of values reported in the literature in populations without known CVD [54,55,56,57].

The association between DD and insulin resistance, as well as a worsening of diastolic function parameters in the presence of metabolic syndrome in patients with or without diabetes was identified by Fontes-Carvalho [58]. In our study, baseline HOMA C-peptide was significantly associated with LAVi in patients with DD. After 12 months, the correlation analysis revealed a significant statistical relationship between LAVi and HbA_1c_ (*p* = 0.018) in patients with DD, as well as between LAVi and index C-Peptide (*p* = 0.04).

Significant associations between long-term glycemic status quantified as HbA_1c_ and parameters of diastolic and systolic dysfunction were reported in multiple studies, where the cardiac function of T2DM patients was evaluated by means of echocardiography have reported [59]. Based on univariate and multivariable analyses, Guria et al. concluded that higher levels of HbA_1c_ increased the risk of LVDD (OR 1.26) [60]. Other researchers found that HbA_1c_ was not associated with LVDD in T2DM overweight/obese patients [61]. It seems, therefore, that the risk of LVDD could be mediated by some other confounding variables, such as BMI. Interestingly, in another study, a strong positive correlation was noted between glycemic variability and E/e’, regardless of which antihyperglycemic medication was administered [62].

In our study, the univariate analysis revealed that a 1% reduction in HbA_1c_ predicted a 32% reduction in the risk of DD. On the other hand, age came through as a risk factor for the development of DD, so it should be accounted for when conducting statistical analyses. In the literature, Bergerot et al. found age, retinopathy, and increased blood pressure over 3 years to be associated with an increased risk of diastolic function deterioration in patients with T2DM [63].

In another study where diabetes patients had long-term glycemic imbalance, despite presenting normal LVEF, reduced values of e’ and global longitudinal strain (GLS) were recorded. The reduction in HbA_1c_ and LDL cholesterol levels was correlated with a relative improvement in GLS, septal e’, and E/e’ after 12 months of intervention. The achievement of statistical significance for the improvement of systolic and diastolic function was observed in patients who had the greatest decrease in HbA_1c_. No significant changes were observed in LA volume, E/A ratio, and LV mass index [64].

In the prospective observational ARIC study, which enrolled 15,744 patients with T2DM and multiple cardiovascular risk factors, the analysis of echocardiographic data for approximately 4400 participants without histories of heart failure or coronary artery disease revealed associations between glycemic status and cardiac structure as well as function. A higher degree of glycemic imbalance was associated with reduced systolic function, increased left ventricular mass, and diastolic dysfunction. The direct relationship highlighted that the E/e’ ratio increased by 0.5 for each percentage point increase in HbA_1c_. These relationships were observed independently of the presence or absence of specific symptoms of heart failure [65].

In a previously mentioned meta-analysis and systematic review, Zhang et al. evaluated the effect of non-insulin antihyperglycemic agents on ventricular remodeling, thereby delaying the onset of heart failure [22]. The authors identified eleven clinical trials that assessed the treatment effects on E/e’, five trials that monitored e’ specifically, and another fourteen trials that analyzed changes in E/A. The results showed that, compared to placebo, GLP-1RAs lead to increased EF, decreased left ventricular telesystolic volume, and reduced E/e’ diastolic index. In our study, for patients treated with exenatide, EF did not change significantly in the active group (67.48 vs. 67.19, *p* > 0.05).

According to a review of cardiovascular outcome trials, the beneficial effect of GLP-1RAs was more significant for patients with both cardiovascular disease and diabetes compared to those with cardiovascular disease but without diabetes. Possible explanations for the mechanisms that cause cardioprotective effects include increased natriuresis, reduced blood pressure values, reduced ischemic injury, stabilization of atherosclerotic plaque, and reduced proliferation of smooth muscle cells. These effects may be the result of the direct action of GLP-1 on the atherosclerosis process and also on the ventricle [66].

A favorable regression of diastolic dysfunction (increased e’ and decreased LV diastolic volume) was observed by magnetic resonance imaging in patients with diabetes without cardiovascular disease under treatment with liraglutide for 6 months, compared to placebo [33]. Similarly, the treatment with daily exenatide reduced the E/e’ diastolic index after 3 months [67]. In our study, the reduction in E/e’ in the active group was not significant. A decrease in the diastolic index was also achieved in the SGLT2 inhibitors group (−1.91) [30].

Literature data regarding the effects of DPP-4 inhibitors on diastolic dysfunction parameters are contradictory. Some researchers report no significant changes, as in the case of vildagliptin, while others observed a negative impact on the telediastolic volume of the left ventricle, or reductions only in the diastolic index [68,69]. According to our results, a significant increase in LVEDD of +5.4 mL was achieved in patients treated with sitagliptin or saxagliptin.

The retrospective analysis of patient subgroups from the PROLOGUE and ASSET trials provides echocardiographic evidence of changes in E/e’ in the group treated with sitagliptin, compared to other conventional medications or to SGLT2 inhibitors [25,70]. In the case of the 55 patients evaluated by echocardiography in the PROLOGUE study, the treatment with sitagliptin was associated with significant modifications of the E/e’ diastolic index after 24 months. The factors included in the multivariate analysis were HbA_1c_, blood pressure, and previous antihyperglycemic medications. Reductions in E/e’ were noted (−0.18) at *p* < 0.05 [25].

In our study, minimal increases of 0.19 for E/e’ and of 0.91 for lateral s’ were observed, but they were not statistically significant. In addition, the mean values of EF, e’, and LAVi slightly decreases, while the telesystolic diameter of the LV, IVS, PWLV, IVRT, and EDT went up, but only the changes in LVEDD and IVRT were statistically significant (*p* = 0.035 and *p* = 0.019, respectively). The number of patients with normal function increased to 24, while the number of patients with DD decreased to 20.

Whether or not cardiovascular medication with demonstrated benefits has the same positive effects on diastolic dysfunction parameters (subclinical myocardial impairment in the context of diabetes mellitus) remains to be demonstrated in longitudinal head-to-head studies. Such studies should feature comparisons between therapeutic classes as well as between different representatives of the same class.

### 3.3. Study Limitations

The relatively small number of patients in the subgroups did not afford certain statistical analyses we would otherwise have been interested in, such as a multivariate analysis of factors predictive of DD, or adjustments for confounding factors such as age or gender. The enrollment of more patients, including adequately controlled and complicated/severe cases, was not feasible at the time of the study. Additionally, more exhaustive exclusion criteria were prioritized in this case to isolate the object of study as much as possible—early diastolic dysfunction in the presence of long-term glycemic imbalance but without atherosclerotic cardiovascular manifestations. Despite these limitations, the research design included consecutive patient enrollment and we were careful to organize the study groups so that baseline characteristics would be statistically homogeneous across them, which compensates for the lack of randomization.

The results should be interpreted knowing that not all confounding factors which could interfere with diastolic function improvement or deterioration were studied in this research. Physical activity, eating behaviors, body composition, and functional capacity were not assessed. Oxidative stress and other inflammatory markers could be included in further research. Regarding follow-up, a monitoring period longer than one year could facilitate a more exhaustive analysis of predictive factors for DD, such as diabetes duration, certain other antihyperglycemic drugs, different outcome cutoffs for glycemic control/imbalance, and concurrent treatments for other known modifiable risk factors (dyslipidemia and hypertension).

The only adverse reaction reported was mild initial dyspepsia in four female patients. There were no cases of patients complaining of nausea, vomiting, or diarrhea during the study period [71]. However, we cannot firmly exclude the possibility of adverse reactions in the three patients who declined the follow-up visit without giving reasons.

Regarding the TNF-α increase, one of the possible explanations can be related to DPP-4 inhibitors treatment. Cases of patients treated with DPP-4 inhibitors complaining of (severe) joint pain have been reported [72]. Indeed, joint pain is facilitated by chronic subclinical inflammation, such as in autoimmune arthropathies [73]. It is possible that some patients had undiagnosed or undisclosed disorders which progressed silently and expressed as increasing levels of TNF-α at follow-up. This illustrates the complexity that characterizes the mechanisms of subclinical inflammation in the presence of obesity and gluco- and lipotoxicity. Testing hypothesized relationships such as between TNF-α and changes in adiposity under each treatment would have required a larger dataset.

We are also aware that our results may challenge trends of cardioprotection through more recently introduced GLP-1RAs and SGLT2 inhibitors. However, in our view, these have not yet been conclusively studied in terms of their effects on DD specifically, which is partly due to the heterogeneity of DD evaluation methods.

Last but not least, from a pragmatic standpoint, not all national health systems and insurance plans provide access to and/or coverage of the full spectrum of approved medication. Romania is among the countries where diabetes treatments are largely paid for by the state (which is very good for the patients), but where innovation is not always readily adopted, so only the more affluent patients can afford the latest drugs. At the same time, the increasing number of patients being diagnosed with diabetes, obesity, etc. adds further pressure on already-strained and underfunded healthcare [74]. In this context, the combination of drugs featured in our study is, concurrently, a limitation as well as a relevant selection in the clinically and economically challenging times ahead.

## 4. Materials and Methods

This was a prospective observational study conducted in a single center over a period of one year, between June 2016 and February 2018, on consecutive patients with T2DM monitored in the Clinical Center for Diabetes, Nutrition and Metabolic Diseases in Iași, NE Romania. The study adhered to the ethical recommendations outlined in the 1975 Helsinki Declaration and received formal approval from both the Research Ethics Committee of the “Grigore T. Popa” University of Medicine and Pharmacy Iași, and the Ethics Committee of the “Sf. Spiridon” Emergency Clinical Hospital, also located in Iași (reference number 63274/16.12.2015).

The inclusion criteria were written informed consent, age between 35 and 80, established diagnosis of T2DM, and HbA_1c_ levels above 7% despite treatment with metformin and/or sulphonylurea or acarbose. These patients required the addition of an incretin-based agent due to glycemic imbalance, in accordance with national protocols.

The main exclusion criteria at the time of telephone contact were the presence of atherosclerotic cardiovascular disease and smoking status. Additionally, patients with insulin-based treatment, type 1 diabetes or secondary pancreatic diabetes, elevated triglycerides above 400 mg/dL, uncontrolled blood pressure above 140/90 mmHg, atherosclerotic cardiovascular disease (e.g., myocardial infarction, angina, coronary revascularization, electrocardiogram findings of ischemia, stroke, transient ischemic attack, and peripheral arterial disease) or valvular heart disease, mitral annular calcification, dysrhythmias, or use of a cardiac pacemaker were excluded from the study. Patients with past medical histories of inflammatory and severe, acute, or chronic conditions (e.g., pancreatitis, liver failure, gastrointestinal and kidney diseases, and malignancies), psychiatric disorders, and pregnancy, or intention to become pregnant, were also excluded, as were patients non-compliant with the research process.

The patients were divided into two groups based on the addition of an incretin agent to their previous treatment regimen; the patients in the active group were started on sitagliptin/saxagliptin (DPP-4 inhibitors available in Romania at the time of enrollment) or on exenatide. At 52 weeks, the participants underwent a comprehensive reassessment of their health status, as follows.

After an overnight fast of 8 h, blood was drawn to evaluate lipid and glycemic levels, as well as kidney and liver function. The blood samples were immediately centrifuged at 3000× *g* for 5 min to ensure accurate measurement of insulin and C-peptide. The serum was then stored at −20 °C and later analyzed using chemiluminescence techniques (IMMULITE 1000) to measure hsCRP, insulin, and C-peptide levels. HbA_1c_ was measured using ion-exchange high-performance liquid chromatography.

A single investigator performed the echocardiographic assessment using Sanoscape SSI 5000. Two-dimensional echocardiography techniques, including color Doppler and pulsed-wave tissue Doppler, were used to determine the parameters for cardiac remodeling and diastolic dysfunction. Based on the American Society of Echocardiography and the European Association of Echocardiography algorithms, the presence and degree of diastolic dysfunction were established using four criteria: septal e’ velocity < 7 cm/s, lateral e’ velocity < 10 cm/sec, E/e’ > 14, LA volume indexed to body surface > 34 mL/m^2^, and TR velocity > 2.8 m/s. A diagnosis of diastolic dysfunction is established when three or four of these conditions are met [75].

The patient data were collected and processed in IBM SPSS Statistics for Windows (version 17, SPSS Inc., Chicago, IL, USA). The descriptive statistical analysis took into consideration frequencies, mean and median values, standard deviation, minimum and maximum values, and interquartile ranges. Depending on the normality of data distribution, comparisons between the study groups were made based on either mean or median values (parametric t-test and non-parametric Mann–Whitney U test). Similarly, correlation formulae were used appropriately to identify significant associations between sets of variables (Pearson and Spearman), where *p* < 0.05. The Wilcoxon signed-rank test was used to compare ranks. Univariate analyses were also performed to assess which initial characteristics and biomarker levels were risk factors for diastolic dysfunction.

## 5. Conclusions

This study evaluated the 1-year outcomes of incretin-based treatment in uncontrolled type 2 diabetes mellitus without clinical manifestations of atherosclerotic cardiovascular disease, with particular focus on diastolic (dys)function parameters. At baseline, half of the patients presented grade 1 LVDD according to the latest guideline recommendations. After 12 months of exenatide or sitagliptin/saxagliptin added to metformin, improved metabolic profile and decreased hsCRP levels were among the significant benefits. Modest significant improvements in LV end diastolic and systolic diameters and IVRT were noted in the group of patients receiving the enhanced treatment regimen. Additionally, diastolic dysfunction regressed favorably for some of these patients, but the differences compared to patients under standard metformin treatment were not significant.

These cardiometabolic benefits are likely the cumulated direct and indirect effects of the studied incretin-based treatment on hyperglycemia, insulin resistance, lipotoxicity, and inflammation, all of which contribute to early diastolic dysfunction. The research is limited in that oxidative stress and altered calcium handling were not considered, and that certain confounding factors could not be controlled in the multivariate analysis (age, gender, diabetes duration, and related complications).

The results reinforce the message that early echocardiographic assessment is necessary for patients with diabetes. The subclinical progression of diastolic dysfunction should not be underestimated, and a cautious, multidisciplinary therapeutic plan is the appropriate approach to prevent the progression to heart failure and the risk of (major) adverse cardiovascular events.

## Figures and Tables

**Figure 1 diagnostics-13-02817-f001:**
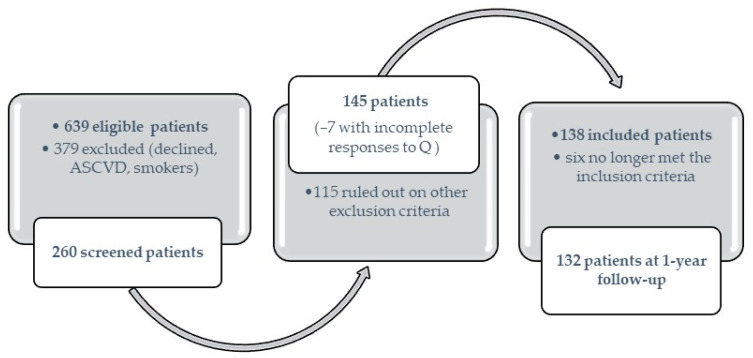
Patient enrollment and inclusion/exclusion flowchart.

**Figure 2 diagnostics-13-02817-f002:**
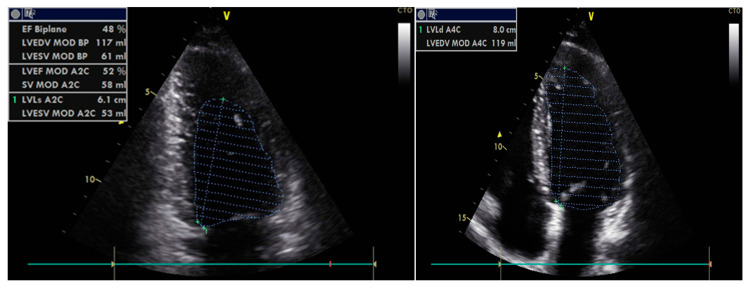
Bidimentional thoracic echocardiography: assessing the ejection fraction of the left ventricle by the Simpson method.

**Figure 3 diagnostics-13-02817-f003:**
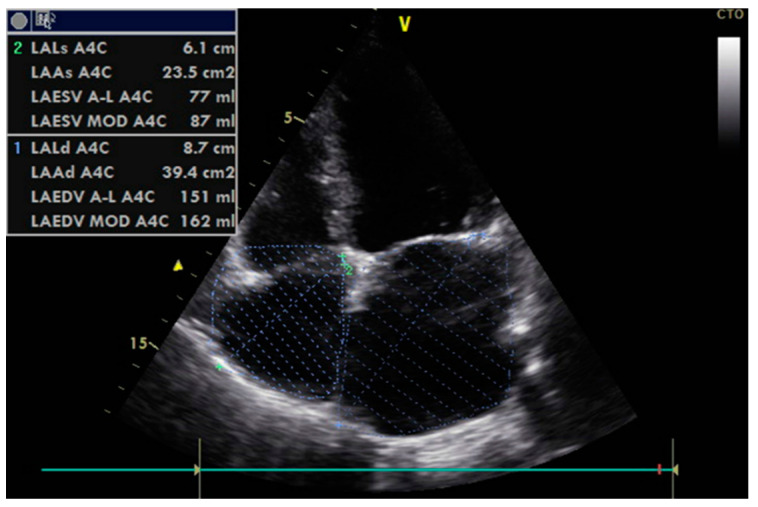
Bidimensional thoracic echocardiography: assessing the size and volume of the left and right atrium.

**Table 1 diagnostics-13-02817-t001:** Characteristics of the study population at baseline (T0) and 12-month follow-up (T1).

Studied Variables		Incretin Treatment Group	Control Group	*p*-Value
Diabetes duration * (years)		5.5 (7)	4.5 (9)	0.470
Age (years)	T0	57.12 ± 9.28	59.33 ± 7.27	0.167
	T1	58.11 ± 9.29	60.33 ± 7.77	0.179
	Δ	0.98 ± 0.18, *p* < 0.001 **	1 ± 0.00, *p* < 0.001 **	0.691
BMI (kg/m^2^)	T0	33.19 ± 5.91	31.65 ± 4.51	0.090
	T1	34.01 ± 15.75	31.15 ± 4.70	0.246
	Δ	0.89 ± 15.55, *p* = 0.591	−0.5 ± 1.98, *p* = 0.105	0.560
Waist circumference (cm)	T0	110.30 ± 10.81	108.64 ± 10.22	0.167
	T1	108.10 ± 9.82	107.45 ± 9.84	0.725
	Δ	−2.19 ± 5.37, *p* < 0.001 **	−1.19 ± 5.25, *p* = 0.150	0.318
HbA_1c_ * (%)	T0	7.8 (1.08)	7.55 (1.40)	0.039 **
	T1	7.2 (1.10)	7.2 (1.10)	0.936
	Δ	−0.66 (1.30), *p* < 0.001 **	−0.45 (1.23), *p* = 0.001 **	0.262
Fasting glycemia * (mg/dL)	T0	165 (46)	162 (50)	0.285
	T1	148 (42)	142.50 (41)	0.599
	Δ	−14 (53.75), *p* < 0.001 **	−9 (42.25), *p* = 0.07	0.310
Insulin * (µIU/mL)	T0	13.85 (10.57)	9.45 (7.44)	0.021 **
	T1	11.10 (9.82)	9.2 (7.05)	0.296
	Δ	−1.46 (10.57), *p* = 0.017 **	−0.59 (5.65), *p* = 0.337	0.511
C-peptide (ng/mL)	T0	3.35 ± 1.45	3.07 ± 1.32	0.236
	T1	3.42 ± 1.67	2.44 ± 1.21	0.001 **
	Δ	0.059 ± 1.97, *p* = 0.777	−0.64 ± 1.77, *p* = 0.022 **	0.050
HOMA-IR	T0	6.26 ± 4.18	4.60 ± 2.95	0.014 **
	T1	4.40 ± 3.26	3.48 ±2.26	0.089
	Δ	−1.86 ± 4.04, *p* < 0.001 **	−1.12 ± 3.44, *p* = 0.031 **	0.292
HOMA C-peptide	T0	4.24 ± 2.03	3.60 ± 2.22	0.091
	T1	3.54 ± 2.01	2.42 ± 1.47	0.001 **
	Δ	−0.70 ± 2.58, *p* = 0.011 **	−1.17 ± 2.83, *p* = 0.007 **	0.332
Index C-peptide *	T0	0.23 (0.16)	0.28 (0.30)	0.003 **
	T1	0.24 (0.18)	0.32 (0.32)	0.004 **
	Δ	0.01 (0.22), *p* = 0.306	0.004 (0.34), *p* = 0.507	1.000
Total cholesterol (mg/dL)	T0	192.22 ± 45.06	197.95 ± 44.91	0.721
	T1	191.02 ± 44.22	184.60 ± 41.77	0.429
	Δ	−1.19 ± 45.43, *p* = 0.805	−13.34 ± 43.81, *p* = 0.052	0.148
LDL cholesterol (mg/dL)	T0	102.72 ± 38.58	102.63 ± 38.73	0.623
	T1	95.13 ± 35.63	91.23 ± 35.99	0.576
	Δ	−7.58 ± 38.41, *p* = 0.087	−11.40 ± 42.86, *p* = 0.100	0.626
HDL cholesterol (mg/dL)	T0	55.51 ± 15.01	60.23 ± 16.24	0.130
	T1	57.17 ± 14.77	55.63 ± 13.82	0.568
	Δ	1.65 ± 13.57, *p* = 0.258	−4.60 ± 14.27, *p* = 0.041 **	0.016 **
Triglycerides (mg/dL)	T0	205.61 ± 95.34	183.21 ± 69.06	0.167
	T1	191.63 ± 84.81	184.60 ± 78.98	0.650
	Δ	−13.98 ± 84.41, *p* = 0.124	1.39 ± 71.41, *p* = 0.899	0.306
eGFR (mL/min/1.73 m^2^)	T0	83.84 ± 18.14	78.39 ± 16.71	0.106
	T1	77.10 ± 18.24	77.56 ± 16.58	0.892
	Δ	−6.73 ± 11.60, *p* = 0.001 **	−0.82 ± 13.00, *p* = 0.685	0.011 **
UACR (mg/g)	T0	27.10 ± 47.01	23.52 ± 16.58	0.559
	T1	16.02 ± 20.95	12.20 ± 47.84	0.993
	Δ	−11.07 ± 31.88, *p* = 0.009 **	−11.31 ± 43.18, *p* = 0.148	0.978
Uric acid (mg/dL)	T0	5.28 ± 1.34	5.70 ± 1.48	0.387
	T1	5.28 ± 1.34	5.66 ± 1.07	0.106
	Δ	−0.10 ± 1.07, *p* = 0.355	−0.03 ± 1.46, *p* = 0.876	0.752
hsCRP * (mg/L)	T0	5.33 (8.24)	5.7 (12.09)	0.983
	T1	3.57 (4.98)	5.96 (8.93)	0.196
	Δ	−1.33 (4.91), *p* = 0.001 **	0 (10.77)	0.238
IL-6 (pg/mL)	T0	2.98 ± 1.90	3.50 ± 2.89	0.089
	T1	3.55 ± 2.43	4.23 ± 2.39	0.102
	Δ	0.56 ± 2.81, *p* = 0.061	0.72 ± 3.89, *p* = 0.237	0.816
TNF-α (pg/mL)	T0	8.12 ± 5.63	7.41 ± 3.30	0.578
	T1	9.56 ± 6.92	8.57 ± 3.15	0.585
	Δ	1.43 ± 3.61, *p* = 0.001 **	0.36 ± 3.93, *p* = 0.547	0.125

* Data are expressed as medians and IQR (non-normal distribution); Δ: the difference between T1 and T0; **: Statistical significance; BMI: body mass index; HbA_1c_: glycated hemoglobin; HOMA-IR: homeostatic model assessment of insulin resistance; HOMA C-peptide: homeostatic assessment model of C-peptide; eGFR: estimated glomerular filtration rate; UACR: albumin-to-creatinine ratio in urine; hsCRP: high-sensitive C-reactive protein; IL-6: interleukin 6; TNF-α: tumor necrosis factor-alpha.

**Table 2 diagnostics-13-02817-t002:** Echocardiographic parameters at baseline (T0) and 12-month follow-up (T1).

Studied Variables		Incretin Treatment Group	Control Group	*p*-Value
LAVi (mL/m^2^)	T0	43.40 ± 11.57	45.15 ± 12.42	0.595
	T1	43.32 ± 10.55	45.49 ± 12.55	0.496
	Δ	−0.082 ± 8.80, *p* = 0.937	0.341 ± 8.53, *p* = 0.799	0.804
LA area (cm^2^)	T0	24.49 ± 4.34	24.14 ± 4.34	0.605
	T1	24.15 ± 4.19	24.23 ± 4.50	0.913
	Δ	−0.34 ± 3.07, *p* = 0.288	0.09 ± 2.55, *p* = 0.812	0.417
E	T0	51.92 ± 12.19	50.95 ± 11.01	0.686
	T1	53.38 ± 11.15	52.91 ± 11.42	0.820
	Δ	1.46 ± 9.94, *p* = 0.170	1.95 ± 8.08, *p* = 0.121	0.778
A	T0	53.01 ± 17.88	50.96 24.76	0.680
	T1	55.01 ± 19.12	54.91 24.11	0.983
	Δ	2.00 ± 10.98, *p* = 0.072	3.96 ± 12.29, *p* = 0.118	0.401
E/A	T0	1.09 ± 0.44	1.13 0.53	0.249
	T1	1.07 ± 0.40	1.17 0.53	0.675
	Δ	−0.02 ± 0.32, *p* = 0.496	0.03 ± 0.25, *p* = 0.395	0.276
EDT	T0	192.05 ± 42.31	198.17 ± 44.81	0.363
	T1	199.83 ± 44.45	206.19 ± 41.90	0.442
	Δ	7.77 ± 37.83, *p* = 0.060	8.02 ± 25.13, *p* = 0.045 **	0.965
IVRT (ms)	T0	102.99 ± 18.81	105.77 ± 18.53	0.642
	T1	106.97 ± 18.74	109.30 ± 21.17	0.141
	Δ	3.97 ± 18.76, *p* = 0.049 **	3.53 ± 15.45, *p* = 0.141	0.894
Lateral e’	T0	9.20 ± 2.53	9.14 ± 2.50	0.922
	T1	9.35 ± 2.43	9.14 ± 2.70	0.756
	Δ	0.15 ± 0.92, *p* = 0.468	0 ± 1.58, *p* = 1	0.622
Septal e’	T0	7.45 ± 1.72	7.51 ± 1.48	0.456
	T1	7.09 ± 1.66	7.33 ± 1.79	0.649
	Δ	−0.35 ± 1.55, *p* = 0.032 **	−0.18 ± 0.32, *p* = 0.376	0.522
Mean e’	T0	8.27 ± 1.96	8.30 ± 1.70	0.964
	T1	8.29 ± 1.80	8.24 ± 1.95	0.928
	Δ	0.01 ± 1.57, *p* = 0.931	−0.06 ± 1.23, *p* = 0.750	0.784
E/e’ *	T0	6.40 (2.41)	6.17 (2.06)	0.274
	T1	6.29 (2.14)	6.47 (1.38)	0.656
	Δ	0.20 (1.51), *p* = 0.308	0.37 (0.86), *p* = 0.019 **	0.442
Lateral s’ *	T0	6.55 (1.4)	6.8 (1.8)	0.493
	T1	6.95 (1.8)	7 (1.8)	0.697
	Δ	0.4 (0.95), *p* = 0.07	0.2 (0.4), *p* = 0.02 **	0.537
IVS	T0	11.45 ± 1.68	11.19 ± 1.57	0.477
	T1	11.55 ± 1.55	11.55 ± 1.46	0.996
	Δ	0.09 ± 1.58, *p* = 0.572	0.35 ± 0.97, *p* = 0.021 **	0.245
LVPW	T0	11.01 ± 1.52	10.93 ± 1.35	0.935
	T1	11.13 ± 1.53	11.36 ± 1.48	0.424
	Δ	0.12 ± 1.50, *p* = 0.441	0.43 ± 0.87, *p* = 0.002 **	0.143
LV ejection fraction (%)	T0	67.48 ± 9.56	66.46 ± 8.97	0.547
	T1	67.19 ± 7.06	65.58 ± 7.25	0.226
	Δ	−0.65 ± 8.37, *p* = 0.462	−0.76 ± 7.48, *p* = 0.505	0.943
LV fractional shortening (%)	T0	38.57 ± 8.08	37.43 ± 7.88	0.480
	T1	38.37 ± 6.79	37.10 ± 7.65	0.489
	Δ	−0.20 ± 6.90, *p* = 0.789	−0.32 ± 5.46, *p* = 0.709	0.922
LVEDD (mm) *	T0	47 (10)	50 (9)	0.030 **
	T1	49 (11)	51 (8)	0.119
	Δ	2 (3), *p* = 0.05 **	0 (1), *p* = 0.09	0.984
LVESD (mm)	T0	27.84 ± 7.53	29.44 ± 8.34	0.293
	T1	29.48 ± 8.12	29.24 ± 8.17	0.969
	Δ	1.64 ± 6.41, *p* = 0.022 **	−0.19 ± 6.54, *p* = 0.850	0.140

* Data are expressed as medians and IQR (abnormal distribution); Δ: the difference between T1 and T0; **: Statistical significance; LAVi: indexed left atrium volume; LA: left atrium; E: mitral E wave velocity (rapid filling) with pulsed Doppler; A: mitral A wave velocity (atrial contraction) with pulsed Doppler; EDT: E-wave deceleration time; IVRT: isovolumic relaxation time; e’: mitral annular velocity with tissue Doppler imaging; s’: systolic velocity; IVS: interventricular septum; LV: left ventricular; LVPW: left ventricular posterior wall; LVEDD: LV end-diastolic diameter; LVESD: LV end-systolic diameter.

**Table 3 diagnostics-13-02817-t003:** Statistical associations between diastolic (dys)function and other patient characteristics and results at baseline.

LV Diastolic Dysfunction	DD Parameters	Age	Disease Duration	BMI	WC	HbA_1c_	HOMA-IR	HOMA C Peptide	Index C-Peptide	UACR	TNF-α	IL-6	hsCRP
Absent		−0.135	−0.077	0.373 *	0.209	−0.194	0.408 **	0.352 *	−0.224	0.136	0.094	0.201	0.214
Present	E	−0.142	−0.103	0.196	0.145	−0.086	0.050	−0.125	0.124	0.061	0.121	0.164	0.002
Indeterminate		0.218	−0.033	0.464	0.471	0.101	0.110	0.159	−0.309	0.107	0.300	0.413	0.249
Absent		−0.262	−0.063	0.292	0.289	−0.184	0.161	0.213	−0.285	0.356 *	−0.112	−0.039	0.093
Present	E/A	−0.367 **	−0.361 **	−0.007	0.013	−0.168	0.087	−0.070	0.174	−0.027	−0.070	0.233 *	−0.010
Indeterminate		−0.330	0.377	0.225	0.011	0.101	−0.492	−0.498	0.536 *	0.216	−0.252	0.242	0.249
Absent		−0.249	−0.109	0.237	0.355 *	0.030	0.226	0.299	−0.291	0.119	−0.014	0.101	0.204
Present	e’	−0.240 *	−0.340 **	−0.180	−0.108	−0.203	−0.126	−0.132	0.133	−0.005	−0.049	0.084	−0.012
Indeterminate		−0.765 **	−0.175	0.183	−0.083	−0.199	−0.161	−0.240	0.240	−0.771	−0.186	0.065	0.214
Absent		0.033	0.042	0.030	−0.075	−0.038	−0.007	0.014	−0.012	0.159	−0.020	0.195	−0.015
Present	E/e’	0.139	0.255 *	0.145	0.144	0.004	−0.086	−0.100	0.100	0.027	0.075	0.036	−0.088
Indeterminate		0.796 **	0.258	0.187	0.326	0.251	0.125	0.284	−0.284	0.886 *	0.484	0.066	−0.042
Absent		0.189	−0.245	−0.029	0.078	0.000	−0.254	−0.160	0.149	0.096	0.129	−0.050	−0.237
Present	EDT	0.322 **	0.236 *	−0.041	0.075	0.238 *	−0.009	0.105	−0.155	0.095	0.082	−0.041	−0.013
Indeterminate		0.373	0.399	−0.437	−0.371	0.214	−0.402	−0.339	0.408	0.240	−0.122	−0.183	−0.622 *
Absent		−0.132	0.026	0.043	0.051	0.020	0.097	0.195	−0.154	0.135	0.125	0.097	−0.054
Present	IVRT	−0.036	0.107	0.071	0.112	−0.084	0.142	0.050	−0.045	−0.044	−0.099	0.123	0.090
Indeterminate		0.482	−0.079	−0.003	0.079	0.346	0.032	0.298	−0.263	0.326	−0.035	0.508	0.047
Absent		0.181	0.217	0.036	−0.019	−0.169	0.158	0.095	−0.115	0.213	−0.145	−0.021	0.070
Present	LAVi	0.170	−0.022	0.185	0.174	−0.194	−0.100	−0.249 *	0.224 *	−0.115	−0.022	−0.053	−0.129

* Statistically significant associations where *p* < 0.05. ** Statistically significant associations where *p* < 0.001. HbA_1c_: glycated hemoglobin; HOMA-IR: homeostatic model assessment of insulin resistance; HOMA C-peptide: homeostatic assessment model of C-peptide; hsCRP: high-sensitive C-reactive protein.

## Data Availability

Readers interested in finding out more about the enrollment criteria, clinical investigations, sample processing techniques, and other studied aspects (e.g., self-reported quality of life) can reach out to the corresponding author or refer to Grigorescu et al., 2021 and 2022 [76,77].

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
