# Peer review of "Effects of Incretin-Based Treatment on the Diastolic (Dys)Function in Patients with Uncontrolled Type 2 Diabetes Mellitus: A Prospective Study with 1-Year Follow-Up"

_diagnostics, 2023, doi:10.3390/diagnostics13172817_

Round 1
Reviewer 1 Report
The researchers measured left ventricular diastolic dysfunction using echocardiographic assessments in patients with diabetes who were treated with either exenatide or sitagliptin/saxagliptin added to standard treatment. One issue with this paper involves the research design; no randomization was used to assign patients to treatment and control groups, increasing risk of selection bias. See attached pdf for further comments.
1. What is the main question addressed by the research? The main question addresses the effects of incretin-based treatment on diastolic dysfunction in type 2 diabetes.2. Do you consider the topic original or relevant in the field? Does it
address a specific gap in the field? The research provides new information on the treatment benefits of diastolic dysfunction with incretin drugs and recommends early assessment with echocardiographic assessment.
3. What does it add to the subject area compared with other published
material? Added findings include benefits for diastolic dysfunction, metabolic profiles, and inflammatory profiles from the specific use of the drugs sitagliptin/saxagliptin and exenatide.
4. What specific improvements should the authors consider regarding the
methodology? What further controls should be considered? I have already mentioned that the methodology lacks randomization to control for selection bias when assigning patients to treatment and control groups,, and I suggested that the researchers describe potential uncontrolled confounding factors that may bias the results.I also suggested that the authors describe the effects of exenatide on prolonging digestion and increasing satiety so that patients consume less food.
5. Are the conclusions consistent with the evidence and arguments presented
and do they address the main question posed? The conclusions can be strengthened by suggesting examples of how lack of randomization may have biased the results.
6. Are the references appropriate? The references are appropriate.
7. Please include any additional comments on the tables and figures.
Abbreviations in the tables are defined to describe technical terms. I also commented where abbreviations in the text need defining.

Reviewer 2 Report
The authors propose an interesting work, but some aspects need to be improved:
Diabetes has an etiopathogenesis that is strongly influenced by a lack of physical activity and unhealthy eating habits:
- This should be emphasized in the introduction, emphasizing how any pharmacological treatment cannot be separated from physical activity and a nutritional program (for example 10.14814/phy2.15740, 10.3390/ijerph191610429, 10.3390/nu15153368, 10.2174/22115366086661811260939 03)
- Neither the level of physical activity (which also influences cardiovascular function) nor eating habits were evaluated in the study; this, if it is not possible to add, must be underlined within the limits
The authors should propose a mechanism that can explain the results found.
Although not statistically significant, an increase in TNF is noted. How could it be explained in the treated group?
It needs revision.
Round 2
Reviewer 1 Report
Well done! My comments were addressed. The revision provides a much more balanced perspective of the research.
Reviewer 2 Report
The authors modified the manuscript following my suggestions, improving the quality. So it can be considered for publication.
It needs some revisions.